# Insight into the Growth Mechanism of Mixed Phase CZTS and the Photocatalytic Performance

**DOI:** 10.3390/nano12091439

**Published:** 2022-04-23

**Authors:** Ying Yang, Yaya Ding, Jingyu Zhang, Nina Liang, Lizhen Long, Jun Liu

**Affiliations:** College of Physics Science and Technology & Guangxi Key Laboratory of Nuclear Physics and Technology, Guangxi Normal University, Guilin 541004, China; yangying20210618@163.com (Y.Y.); dyy1622510316@126.com (Y.D.); zhangjy19961023@163.com (J.Z.); liangnina3@163.com (N.L.)

**Keywords:** CZTS, wurtzite, kesterite, growth mechanism, photocatalytic performance

## Abstract

In this work, CZTS particles with a mixed phase of wurtzite and kesterite were synthesized by the solvothermal method. The time-dependent XRD patterns, Raman spectra, SEM, and EDS analysis were employed to study the growth mechanism of CZTS. The results revealed that the formation of CZTS started from the nucleation of monoclinic Cu_7_S_4_ seeds, followed by the successive incorporation of Zn^2+^ and Sn^4+^ ions. Additionally, the diffusion of Zn^2+^ into Cu_7_S_4_ crystal lattice is much faster than that of Sn^4+^. With increasing time, CZTS undergoes a phase transformation from metastable wurtzite to steady kesterite. The morphology of CZTS tends to change from spherical-like to flower-like architecture. The mixed-phase CZTS with a bandgap of 1.5 eV exhibited strong visible light absorption, good capability for photoelectric conversion, and suitable band alignment, which makes it capable to produce H_2_ production and degrade RhB under simulated solar illumination.

## 1. Introduction

Cu_2_ZnSnS_4_ (CZTS), a quaternary chalcogenide p-type semiconductor with a direct bandgap energy of about 1.4~1.6 eV [1], has attracted much attention in the field of solar cell [2,3,4], photoelectrocatalytic [5,6,7,8] and photocatalytic application [9,10,11], due to its high absorption coefficient (>10^4^ cm^−1^), natural abundance and non-toxicity. CZTS can exist in three typical crystal structures, known as kesterite, stannite, and wurtzite [12,13]. Kesterite is the ground state of CZTS, which has a tetragonal supercell derived from cubic zinc-blende lattice. Stannite differs from kesterite only in the arrangements of Cu and Zn atoms. Wurtzite is a metastable CZTS, which possesses a hexagonal crystal cell. Various strategies have been employed to prepare CZTS, including electrodeposition [5,14], sol–gel [15], chemical bath deposition [16], successive ionic layer absorption and reaction [17], and solvothermal synthesis [18,19]. The solvothermal method has been widely adopted to synthesize CZTS particles by the merit of being convenient for manipulation, low-cost, and more suitable for large-scale production. The crystal structures and morphology can be readily controlled by adjusting reaction solvent, sulfur source, metallic precursors, reaction temperature and time, and surfactant concentration [20]. For example, kesterite CZTS is reported to be obtained by using ethylene glycol (EG) [20] and ethylene alcohol [21] as solvents, while wurtzite CZTS is often obtained by using oleylamine (OAm) [1,22], and ethanediamine [23] as solvents. CZTS nanoparticles with kesterite, wurtzite, as well as a mixed both phase structures were obtained by using sulfur, 1-dodecanethiol, and thioacetamide as sulfur sources, respectively [24].

However, a big challenge needs to be solved, which is the formation of binary and ternary sulfides byproducts in the preparation of CZTS. These byproducts are detrimental to the photoelectrical performance of CZTS, leading to the short circuit of solar cell devices and the recombination of electron-hole pairs [25]. Therefore, it is significant to understand the formation pathway of CZTS to achieve a controlled synthetic route for pure-phase CZTS nanoparticles. Up until now, the formation mechanism of the CZTS has not been clear due to the complex and variable reaction conditions. Li et al. considered that the formation of wurtzite CZTS started with the nucleation of Cu_2_S followed by the incorporation of Zn^2+^ and Sn^4+^ into the Cu_2_S crystal lattice and replaced parts of Cu ^+^ [1]. Regulacio et al. [26] reported that djurleite Cu_1.94_S seeds were formed using long-chain alkanethiols organic surfactant, which result in the formation of wurtzite CZTS, while digenite Cu_1.8_S seeds were formed using alkylamines organic surfactant, which lead to the formation of tetragonal CZTS. Ling et al. reported that the growth of wurtzite CZTS starts from the nucleation of Cu_2__-x_S nanoparticles, followed by diffusion of Sn^4+^ and Zn^2+^ into Cu_2__-x_S nanoparticles successively [25]. Conversely, Lu et al. [18,19] reported that the ionic radius of Zn^2+^ (74 pm) was closer to Cu^+^ (77 pm) than Sn^4+^ (69 pm), thus Zn^2+^ could diffuse into the Cu_2_S crystal lattice faster than that of Sn^4+^.

In this work, the mixed phase of CZTS nanoparticles was prepared by the solvothermal method using monoethanolamine as solvent. The time-dependent experiment was carried out to study the growth mechanism. We found that the formation of CZTS started with the nucleation of Cu_7_S_4_, followed by the incorporation of Zn^2+^ and Sn^4+^ successively. The obtained CZTS exhibited the capability to produce hydrogen from water splitting and degraded RhB under simulated solar illumination due to the excellent visible light absorption, good capability for photoelectric conversion, and suitable band alignment.

## 2. Experimental Section

### 2.1. Chemical and Reagents

All the reagents are analytical grade and used without further purification. Copper (II) acetate monohydrate (Cu(CH_3_COO)_2_·H_2_O, ≥99.0%), monoethanolamine (OH(CH_2_)_2_NH_2_, ≥99.0%), tin (IV) chloride pentahydrate (SnCl_4_·5H_2_O, ≥99.0%) and thiourea (H_2_NCSNH_2_, ≥99.0%) were bought from Xilong Scientific Co., Ltd. (Guangdong, China). Zinc (II) acetate dihydrate (Zn(CH_3_COO)_2_·2H_2_O, ≥99.0%) was supplied by Aladdin Reagent Co., Ltd. (Shanghai, China). Anhydrous ethanol (C_2_H_5_OH) was purchased from Cologne Chemicals Co. Ltd. (Chengdu, China). Deionized water was produced using a Direct-Q3 water purification system.

### 2.2. Synthesis of CZTS

The CZTS was prepared by a simple solvothermal method using monoethanolamine as solvent [27]. In a typical synthetic process, copper acetate (2 mmol), zinc acetate (1 mmol), tin chloride (1 mmol), and thiourea (8 mmol) were dissolved sequentially into 64 mL of monoethanolamine under magnetic stirring at room temperature. Then, the precursor was transferred into a Teflon-lined autoclave of 80 mL, and placed in the oven (200 °C). The autoclaves were maintained for different times (0.5, 1, 2, 3, 5, 8, 12, and 24 h) to study the growth mechanism. After the reaction, the black precipitates were washed with deionized water and ethanol several times. The CZTS samples were obtained after drying at 60 °C for 24 h.

### 2.3. Characterizations

The composition and crystal structure was analyzed by X-ray diffractometer (XRD, Mini Flex600, Rigaku, Tokyo, Japan) and Raman spectrum (inVia, Renishaw, London, England) with a semiconductor laser excitation at 514 nm. Au film on sample stage was deposited by ion sputtering for SERS measurements. The morphology, elemental constituent, and mapping were characterized by transmission electron microscopy (TEM, JEM-2100F, Tokyo, Japan) and scanning transmission electron microscopy (SEM, Zeiss Sigma 300, Oberkochen, Germany) equipped with EDS. The chemical valence state was detected by X-ray photoelectron spectroscopy (XPS, K-Alphaþ, Thermo Fisher Scientific, Waltham, MA, USA). UV-Vis absorption spectrum was recorded by spectrophotometer (UV-2700, Shimadzu, Shandong, China). Photoelectrochemical measurements were conducted on an electrochemical workstation (CHI 760E) using a three-electrode system. Pt sheet and Ag/AgCl electrodes were used as counter electrodes and reference electrodes, respectively. The photocatalysts were coated on FTO glasses to serve as working electrodes (1.0 × 1.0 cm^2^). 0.1 M Na_2_SO_4_ solution served as the electrolyte, which bubbled with N_2_ for 30 min before each test. The transient photocurrent was obtained at 0.2 V bias under the chopped simulated solar illumination. Electrochemical impedance spectroscopy (EIS) was conducted at 0.2 V bias in the range of 0.01–10^6^ Hz. The Mott–Schottky measurement was performed at 1 KHz using 0.5 M Na_2_SO_4_ solution as electrolyte.

### 2.4. Photocatalytic Performance

Photocatalytic hydrogen evolution was tested in a quartz reactor using 300 W Xenon lamp as light source. 0.01 g of photocatalyst was dispersed into 90 mL solution (containing 0.25 M Na_2_SO_3_ and 0.35 M Na_2_S) by ultrasonic treatment for 15 min. Before simulated solar irradiation, N_2_ was bubbled into the suspension for 30 min to completely remove the dissolved oxygen. The suspension was magnetic stirring during the whole reaction. The evolved gases were extracted by gas injection needle every 15 min and analyzed by gas chromatography (Agilent 7890B). The photocatalytic degradation of RhB was conducted in a customized photoreactor under simulated solar irradiation. 0.05 g of CZTS powder was dispersed in 50 mL of 10 ppm RhB solution by stirring in the dark to achieve the adsorption/desorption equilibrium. Then, the solutions were exposed to simulated solar irradiation under continuous stirring. At given time intervals, 3 mL of RhB solution was taken out and tested by spectrophotometer after removing CZTS nanocrystals. The degradation efficiency of RhB was evaluated based on the Formula (1):(1)$$\eta =\frac{C_{0}-C_{t}}{C_{0}}\times 100\%$$
where *C*_0_ is the concentration of RhB solution before irradiation, *C*_t_ is the concentration of RhB solution after irradiation for *t* min.

## 3. Results and Discussion

The time-dependent XRD patterns and Raman spectra were measured in order to understand the growth mechanism of CZTS, as shown in Figure 1. The XRD pattern of CZTS-0.5 h matches well with the monoclinic Cu_7_S_4_ (JCPDF: No 23-0958), indicating the nucleation and growth of CZTS start with Cu_7_S_4_, which is in accordance with previous reports [22]. When the time increased, the diffraction peaks of Cu_7_S_4_ decreased and almost disappeared at the reaction time of 3 h, while the diffraction peaks of the wurtzite CZTS plane increased with the time from 0.5 to 3 h. For CZTS-3 h, the diffraction peaks can be indexed to wurtzite CZTS, indicating that the Cu_7_S_4_ nucleus completely transformed into wurtzite CZTS in less than 3 h. It is noticeable that the peaks at 28.5°, 47.3°, and 56.2° increased after 3 h, which is the characteristic of kesterite CZTS (JCPDF: No 26-0575). The results indicated that CZTS undergoes a phase transformation from wurtzite to kesterite. However, the weak diffraction peaks at about 31.3° and 46.4° are observed even for the CZTS-24 h sample, which is mainly attributed to Cu_2__-x_S byproducts. The presence of a small amount of Cu_2__-x_S byproducts is also observed by other groups [28].

Since the diffraction peaks of ZnS, Cu_3_SnS_4_, Cu_2_SnS_3_, ZnS, and SnS are overlapping with that of CZTS, Raman spectra were measured to confirm the phase composition of the CZTS. The time-dependent Raman spectra displayed a similar evolution tendency with XRD patterns. As shown in Figure 1b, the peak centered at 468 cm^−1^ is the characteristic peak of Cu_7_S_4_, and the peak intensity decreased from 0 to 3 h. When the reaction time exceeds 2 h, the peak at 328 cm^−1^ starts to appear, which could be attributed to CZTS. With the reaction time increasing, this peak obviously shifted to 323 cm^−1^, which is probably as a result of phase transition from wurtzite to kesterite phase. The surface-enhanced Raman scattering spectroscopy of the CZTS-24 h sample clearly exhibited three peaks located at 298.4 cm^−1^, 332.9 cm^−1,^ and 359.9 cm^−1^, in good agreement with CZTS (Figure 1c). Characteristic peaks from the possible impurities, such as SnS (193 and 224 cm^−1^) [29], ZnS (351 and 278 cm^−1^) [24] and Cu_2_SnS_3_ (298, 356 cm^−1^) [25], Cu_3_SnS_4_ (318 cm^−1^) [25] were not found. The results of the XRD pattern and Raman spectrum revealed that the CZTS-24 h sample was a mixed phase of wurtzite and kesterite.

The morphology evolution of CZTS with increasing the reaction time is examined by SEM images. The images of CZTS-0.5 h show that the Cu_7_S_4_ nucleus is nanoparticles with dozens of nanometers in size, which agglomerated into a sphere-like structure (Figure 2a). For the CZTS-1 h sample, the nanoparticle size was reduced due to the decomposition of Cu_7_S_4_ and the formation of CZTS nanoparticles (Figure 2b). The morphology did not change too much for CZTS samples with the increase in reaction time from 3 to 12 h, except that the agglomeration becomes more obvious (Figure 2c–g). The regular nanosheets were observed from the CZTS-24 sample (Figure 2h), which were cross-connected to form a flower-like structure. It is reported that high temperature is beneficial to forming the flower-like structure of CZTS [9].

TEM and elemental mapping were employed to further examine the morphology and elemental distribution of the CZTS-24 h sample. Figure 3a shows that the CZTS nanoparticles are in the range of several to tens of nanometers. The HRTEM image (Figure 3b) displayed two kinds of lattice arrangements, as marked by a rectangular frame. These regions were amplified to analyze the lattice fringe as shown in Figure 3c,d, respectively. The quadrangle lattice area shows an interplanar spacing of 0.31 nm, which is ascribed to the (112) plane of tetragonal kesterite CZTS. The hexagon lattice area shows the lattice spacing of 0.33 nm, which is corresponding to the (100) plane of hexagonal wurtzite CZTS. The HRTEM results further confirmed the co-existence of kesterite and wurtzite phases in CZTS-24 h. The elemental mapping of the area marked in Figure 4a confirmed that Cu, Zn, Sn, and S elements are homogenously distributed in the CZTS-24 h sample, as shown in Figure 4b–e.

In order to analyze the valence states of Cu, Zn, Sn, and S elements in CZTS-24, XPS analysis was performed. As shown in Figure 5a, the survey spectrum exhibited the strong signals of Cu, Zn, Sn, and S elements. In the Cu 2p spectrum (Figure 5b), two peaks centered at 932.0 and 951.8 eV with a splitting of 19.8 eV is the characteristic of Cu^+^. For the Zn 2p spectrum (Figure 5c), the peaks located at 1022.1 and 1045.2 eV with a peak distance of 23.1 eV are assigned to 2p_3/2_ and 2p_1/2_ of Zn^2+^, respectively. From the Sn 2p spectrum (Figure 5d), the peak at 486.7 and 495.2 eV with a spin-orbit splitting of 8.5 eV are observed, indicating the presence of Sn^4+^. The S 2p spectrum was fitted with two distinct doublets with a splitting of 1.2 eV (Figure 5e). The peak at 161.9 and 163.1 eV is corresponding 2p_3/2_ and 2p_1/2_ of S^2-^ in CZTS [29]. Whereas the peaks at 162.2 and 163.4 eV can be assigned to other sulfide species, such as Cu_2-x_S byproduct as shown in XRD patterns [28]. The XPS results proved that the valence states of Cu, Zn, Sn, and S elements agree well with CZTS nanoparticles. The XPS quantitative analysis showed that the atomic percent of Cu: Zn: Sn: S in the CZTS-24 h sample is about 1.9:1.2:0.9:4.0.

The element compositions of CZTS nanoparticles synthesized at different reaction times were obtained from EDS and listed in Table 1. For the CZTS-0.5 h sample, only Cu and S elements were present with a stoichiometry ratio quite close to Cu_7_S_4_. After 24 h, the Cu/(Zn + Sn) ratios reduced from 2.42 to 1.03, and the Zn/Sn ratio reduced from 3.97 to 1.31. Based on the time-dependent crystal phase, morphology and the elemental composition evolutions, the growth mechanism of CZTS can be described as follows. In the precursor, thiourea (Tu) not only acts as a sulfur source, but also plays the role of complexing agent to complex with metal ions. At the same time, monoethanolamine (EA), a solvent with one –NH_2_ group per molecule, can also bind with metal ions to form an M-EA complex. Since the pH of EA solvent is about 12, thiourea is unstable in an alkaline environment and ready to decompose into H_2_S. Due to the presence of H_2_S, Cu^2+^ ions may partly reduce by S^2−^ and give rise to Cu^+^ ions [30,31]. With the increase in reaction temperature and pressure, Cu-EA and Cu-Tu complexes were firstly thermally decomposed into Cu^+^ and Cu^2+^ ions, which reacted with S^2-^ to form monoclinic Cu_7_S_4_ seeds [32]. In the following, Zn^2+^ and Sn^4+^ ions were successively released from their mental complex and incorporated into the Cu_7_S_4_ crystal lattice to replace part of Cu^+^ and Cu^2+^. As shown in Table 1, the Zn content is much higher than Sn content in the first 1 h, which indicated that the diffusion of Zn^2+^ into the Cu_7_S_4_ crystal lattice is much faster than that of Sn^4+^. The Cu/(Zn + Sn) and Zn/Sn ratio achieve relative stability after 5 h, indicating the formation of CZTS is almost accomplished. With the time prolonged, the CZTS undergoes a phase transformation from metastable wurtzite to steady kesterite. Due to the high surface free energy, the small particles aggregated into spherical-like morphology. After a 24 h reaction, according to the Ostwald ripening theory, the small nanoparticles gradually coalesced to form nanoflakes, as a result, the spherical-like morphology changed into flower-like architecture.

The optical absorption of the CZTS-24 h was studied by UV–Vis spectra. Figure 6a shows that the CZTS-24 h exhibits a broad absorption in the whole visible-light region, indicating excellent visible light absorption. The bandgap energy (*E*_g_) was determined by the Tauc plot based on Formula (2) [6]. Where α, *hv*, and *A* is absorption coefficient, incident photon energy, and constant, respectively. *E_g_* can be obtained from the Tauc plot by extending the tangent segment to the *x*-axis (inset of Figure 6a), which was determined to be 1.50 eV. As shown in the transient photocurrent curve (Figure 6b), the CZTS-24 h exhibited an obvious photocurrent response under simulated solar illumination, indicating the good capability for photoelectric conversion. Additionally, no obvious decrease in photocurrent was observed, indicating good photostability of the CZTS-24 sample.
(2)$$\alpha hv=A{\left(hv-E_{g}\right)}^{\frac{1}{2}}$$
(3)$$\frac{1}{C^{2}}=\frac{-2}{e\varepsilon_{r}\varepsilon_{0}N_{A}A^{2}}\left(E-E_{FB}-\frac{k_{B}T}{e}\right)$$

The band alignment of CZTS was determined by the Mott–Schottky Equation (3) [33]. As shown in Figure 6c, the M–S plot displays a negative slope, which confirmed that the CZTS is a p-type semiconductor. The flat band potential (*E*_FB_) can be obtained by extrapolating the linear portion to the x-axis, which was determined to be 0.65 V (vs. Ag/AgCl). It is generally accepted that the top of the valence band (VB) for a p-type semiconductor is more positive (0–0.3 eV) than *E*_FB_. Here, the difference between VB and *E*_FB_ is set to 0.2 V. Based on the *E*_FB_ and *E*_g_ values, the CB and VB energy level of CZTS is calculated to be 1.05 eV and −0.45 eV (vs. SHE), respectively. As shown in Figure 6d, the band alignment of CZTS meets the requirements of photocatalytic H_2_ production from water splitting with its electrons in CB and the degradation of most of the organic pollutants using its holes in VB [34]. Figure 6e exhibits the time courses of hydrogen evolution on CZTS-24 h photocatalyst under simulated solar and room temperature. The hydrogen evolution rate is 1042.5 µmol g^−1^ h^−1^ with no obvious decline during the 8 h reaction, indicating good stability for photocatalytic hydrogen production. Figure 6f shows the degradation curve of RhB over CZTS-24 h photocatalyst under simulated solar. The concentration of RhB solution continuously declined during the 220 min irradiation. After 220 min irradiation, the RhB photodegradation efficiency of the CZTS-24 h sample was calculated to be 43.3%, which is higher than that of the blank control group (35.3%).

## 4. Conclusions

CZTS particles with a mixed phase of wurtzite and kesterite were obtained by the solvothermal method using monoethanolamine as solvent. The growth mechanism of CZTS can be described as follows. Firstly, monoclinic Cu_7_S_4_ nuclei were formed in the first 30 min. The Cu_7_S_4_ nucleus is dozens of nanometers in size, which aggregated into a sphere-like structure. In the following, Zn^2+^ and Sn^4+^ ions were incorporated into the Cu_7_S_4_ crystal lattice to form wurtzite CZTS nanoparticles. The diffusion of Zn^2+^ ions into the Cu_7_S_4_ crystal lattice is much faster than that of Sn^4+^ ions. Then, with the time prolonged, the CZTS nanoparticles undergo a phase transformation from wurtzite to kesterite. Some of the small CZTS nanoparticles coalesced to form nanoflakes, which cross-connected to form a flower-like structure. The mixed-phase of CZTS exhibits a bandgap of 1.50 eV and broad optical absorption in the whole visible-light region. The excellent visible light absorption, good capability for photoelectric conversion, and suitable band alignment make it capable to produce H_2_ production and degrade RhB under simulated solar irradiation.

## Figures and Tables

**Figure 1 nanomaterials-12-01439-f001:**
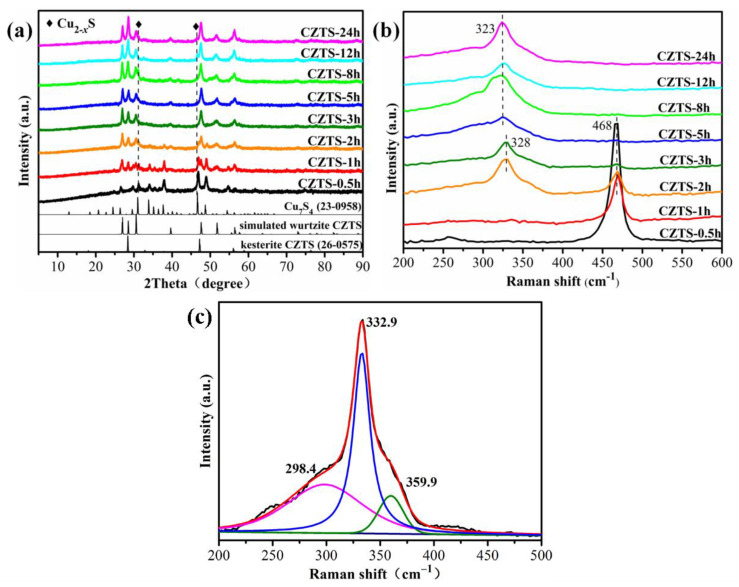
The time-dependent (**a**) XRD patterns and (**b**) Raman spectra of CZTS samples, and (**c**) the SERS of CZTS-24 h sample (black line: real curve; red line: fitted curve; green, blue and magenta: the three splitting peaks of fitted curve).

**Figure 2 nanomaterials-12-01439-f002:**
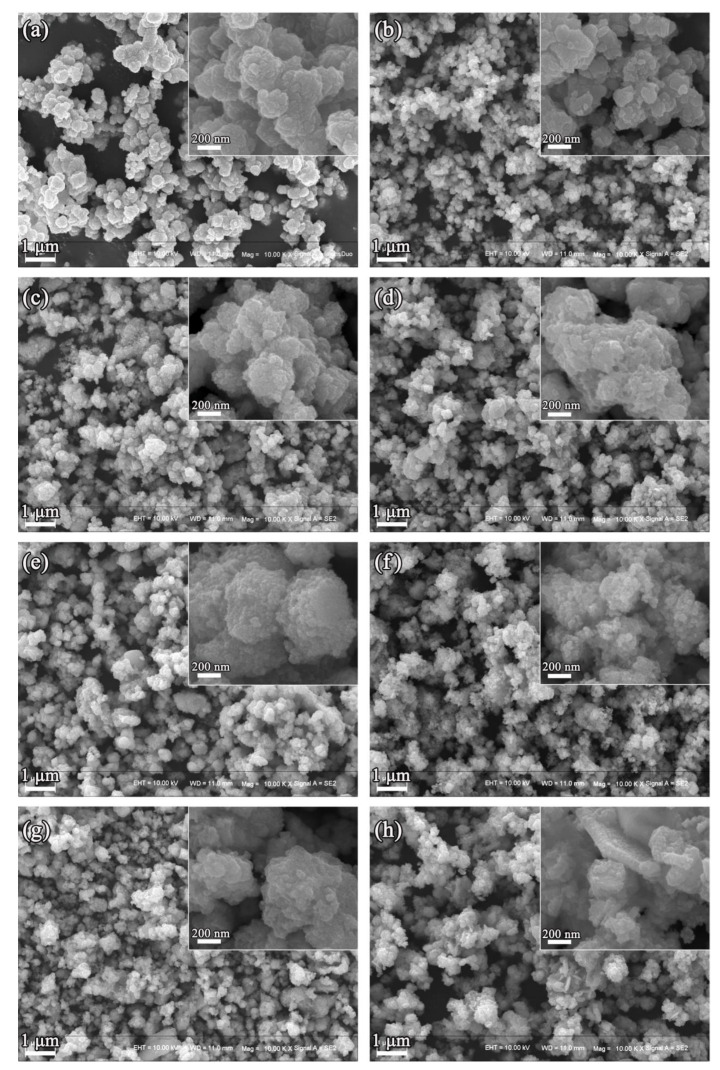
SEM images of CZTS nanoparticles obtained at (**a**) 0.5 h, (**b**) 1 h, (**c**) 2 h, (**d**) 3 h, (**e**) 5 h, (**f**) 8 h, (**g**) 12 h, and (**h**) 24 h.

**Figure 3 nanomaterials-12-01439-f003:**
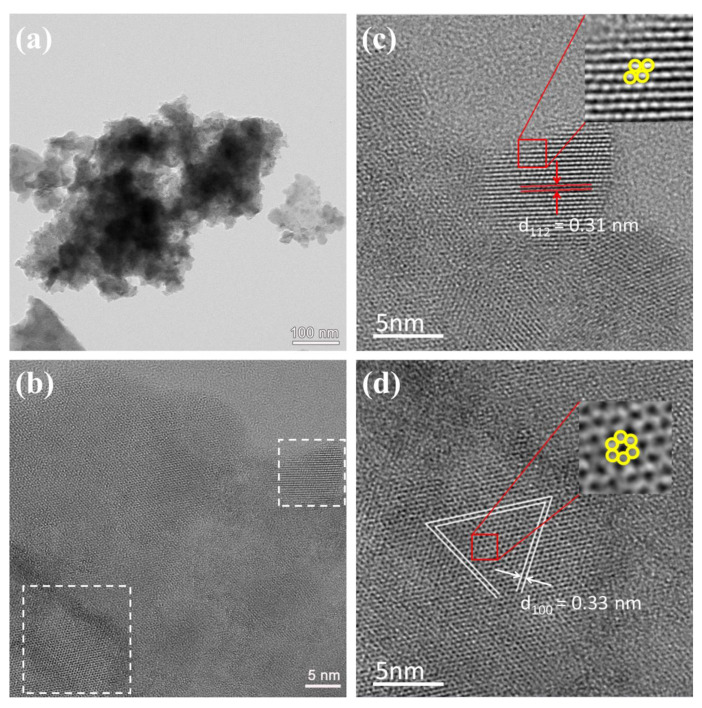
(**a**) TEM and (**b**) HRTEM image of CZTS-24 h sample, (**c**) and (**d**) the magnified images of two regions marked with white rectangles in (**b**).

**Figure 4 nanomaterials-12-01439-f004:**
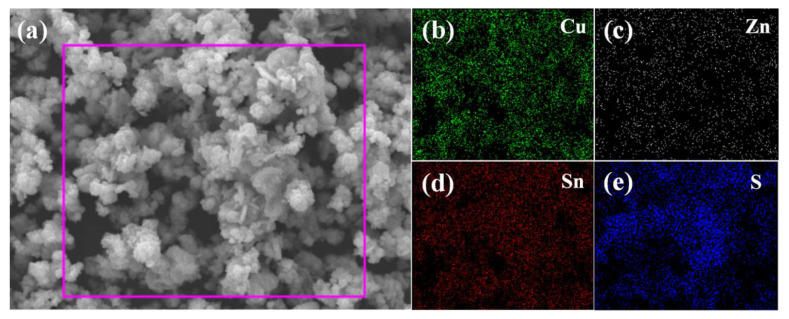
(**a**) SEM image of CZTS-24 h sample and the corresponding elemental mapping of (**b**) Cu, (**c**) Zn, (**d**) Sn, and (**e**) S.

**Figure 5 nanomaterials-12-01439-f005:**
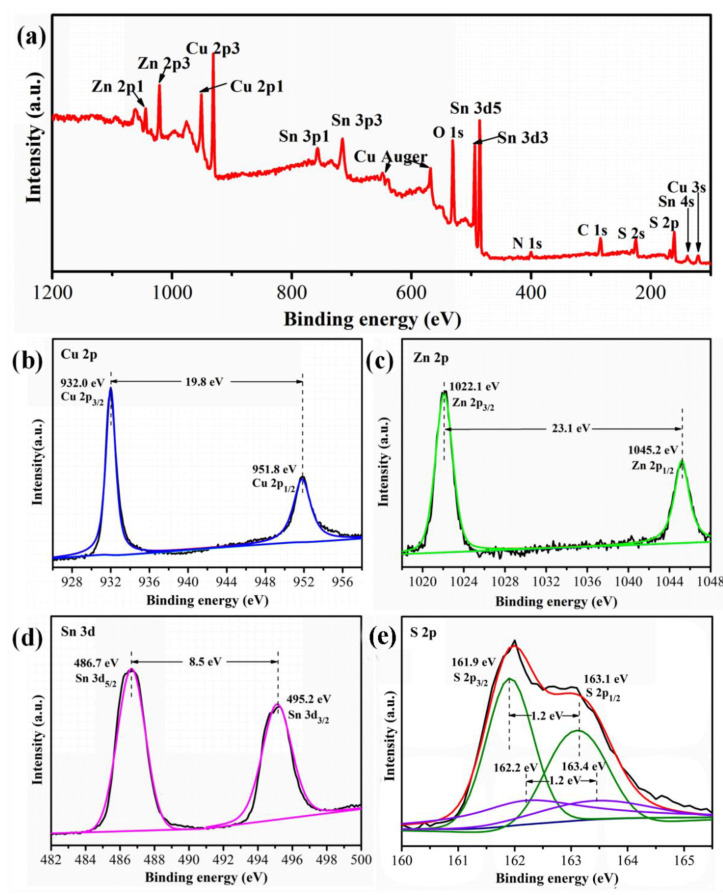
(**a**) XPS survey spectra of CZTS-24 h and the high-resolution XPS spectra of (**b**) Cu 2p, (**c**) Zn 2p, (**d**) Sn 2p and (**e**) S 1s.

**Figure 6 nanomaterials-12-01439-f006:**
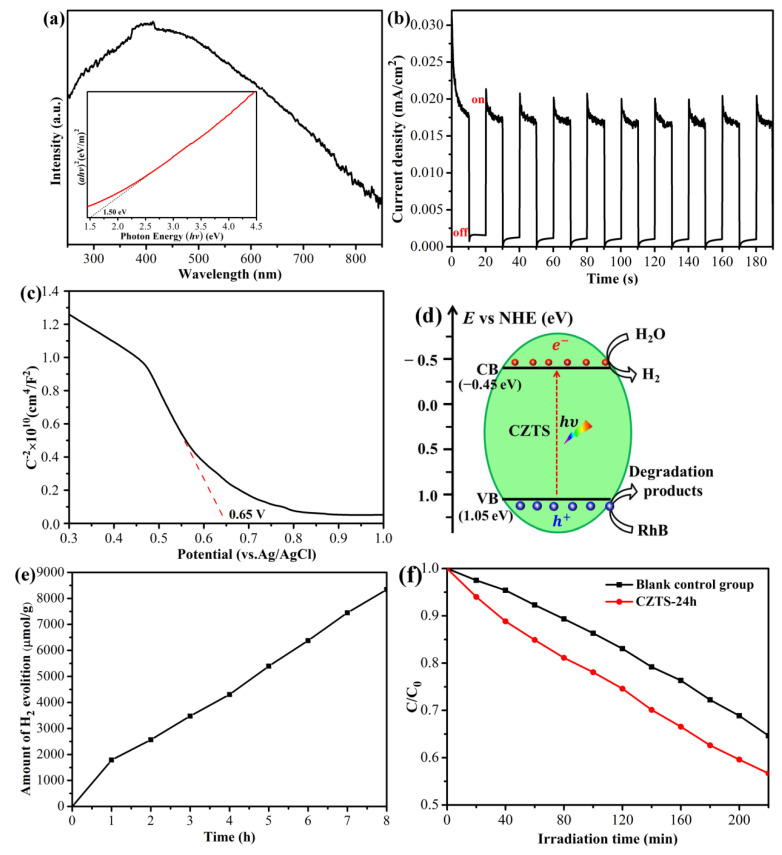
(**a**) UV-Vis absorption spectrum (the inset is the corresponding Tauc plot), (**b**) transient photocurrent curve, (**c**) Mott–Schottky curve, and (**d**) band alignment of CZTS-24 h; (**e**) The course of H_2_ evolution and (**f**) photocatalytic degradation of RhB over CZTS-24 h photocatalyst under simulated solar irradiation.

**Table 1 nanomaterials-12-01439-t001:** Elemental composition of CZTS nanoparticles obtained at different reaction time.

Sample	Composition Ratio					
	Cu	Zn	Sn	S	Cu/(Zn + Sn)	Zn/Sn
**CZTS-0.5 h**	66.33	0	0	33.67	——	——
**CZTS-1 h**	40.72	13.44	3.38	42.47	2.42	3.97
**CZTS-2 h**	42.94	8.04	7.22	41.81	2.81	1.11
**CZTS-5 h**	36.87	13.61	11.53	37.98	1.46	1.18
**CZTS-12 h**	31.96	12.24	10.97	44.83	1.37	1.11
**CZTS-24 h**	26.21	14.47	11.05	48.28	1.03	1.31

## Data Availability

Not applicable.

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
