# Peer review of "Insight into the Growth Mechanism of Mixed Phase CZTS and the Photocatalytic Performance"

_nanomaterials, 2022, doi:10.3390/nano12091439_

Round 1
Reviewer 1 Report
Typo error
- Paragraph 69: monothanolamine --> monoethanolamine
- Paragraph 75: monoethanolamide --> monoethanolamine
- Paragraph 193: 2p3/2 --> 2p5/2 / 2p1/2 --> 2p3/2
- Paragraph 258: monothanolamine --> monoethanolamine
- It was mentioned in the introduction that the problem of binary and ternary sulfide byproduct generated during CZTS formation mentioned in the introduction could be solved by using monoethanolamine solvent. However, it seems that clear experimental results for this part are lacking.
Reviewer 2 Report
Authors developed a mixed phase of wurtzite and kesterite in CZTS for H2 production and RhB degradation under simulated solar environment. They characterized the samples with SEM, but EDS spectra with element compositions are missing. Also additional catalytic performance tests with CZTS-0.5h and CZTS-3h are required for comparing to CZTS-24h, the final mixed phase material. Comments and questions are the following.
1. Abstract: The term “excellent” isn’t proper.
2. Figure 1(a), XRD: Authors analyzed the peak at 28.5o, 47.3o and 56.2o increased after 3 h and concluded CZTS transformed from wurtzite to kesterite. Only the peak at 28.5o is shown well to be stronger, but other two peaks are not seen clearly. It is strongly recommended to include an insert with a magnified region from 45o to 60o for better view. In addition, the wurtzite peaks are still strong, suggesting wurtzite is still the main phase. How do they think about the main phase?
3. Figure 2, SEM: Authors named “flower-like” structure, but it is difficult to agree the name. Another term is required for the structures obtained after 24 h. Please compare Figure 1(c) in Ref. [9] and Figure 1(h) in this manuscript. They look quite different.
4. Figure 3(b), TEM: There are two rectangles. Which one is for (c)? It is recommended to label them with (c) and (d).
5. Page 6, line 172: You analyzed the interplanar spacings of 0.31 and 0.33 nm are ascribed to the (112) plane of tetragonal kesterite CZTS and the (100) plane of hexagonal wurtzite CZTS, respectively. What are the reference literatures?
6. Figure 4, EDS: They already have element mapping data that support only the presence of Cu, Zn, Sn and S elements, so the EDS spectrum with element composition (at. %) should be provided for supporting the ratio of Sn:Zn:Cu:S in CZTS and comparing the atomic composition measured from XPS data in Figure 5 and Table 1. Those are very important characterization results in this manuscript.
7. Figure 5, XPS: It is recommended to cite Ref. [28] for assigning Cu+ at 932.0 and 951.8 eV, Zn2+ at 1022.1 and 1045.2 eV, and Sn4+ at 486.7 and 495.2 eV as you did it for S2- in CZTS.
8. Figure 6(e,f): The photocatalytic performance of CZTS-24h are good, but they need any reference blank data for comparison. What about CZTS-0.5h and CZTS-3h. These are very important for highlighting the mixed phase. They should include their catalytic performance also in Figure 6.
9. Typo: Ref.[9] The title is “The visible light-driven ….”. “The” is missing.
Round 2
Reviewer 2 Report
Authors replied reviewer’s comments and questions and revised the manuscript according to them. Additional comments are the following.
1. The reason for not recommending “excellent” was in preference to accurate evaluation with numbers rather than any non-scientific term like excellent, best or greatest although it is reasonable. Many people mention their samples are best and excellent. It is good, but what if another new material, which has much better performance and properties than them, is reported soon? It would be also excellent, and the previous materials may not be excellent anymore. For example, we can say Moon is really bright. In scientific journals, we like to know how much it is bright in any unit like lux. Reviewer hopes that authors consider this suggestion in that way without any misunderstanding or intention to underestimate.
2. “Flower-like” structure is also in a similar way. There are a variety of flower shape as many flowers. If they cited Ref. 9 just for the term “flower-like”, it would cause any confusion the structure looks like them in Ref. [9]. Authors already agreed their structure is quite different from Ref. [9] in author’s reply, so it is recommended to remove the citation or cite another reference literature that shows a similar flower structure.
